EMBO
Molecular Medicine

# Oligonucleotide-induced alternative splicing of serotonin 2C receptor reduces food intake

Zhaiyi Zhang[1],[†], Manli Shen[1],[†], Paul J Gresch[2], Masoud Ghamari-Langroudi[3], Alexander G Rabchevsky[4], Ronald B Emeson[3] & Stefan Stamm[1],[*]

## Abstract

The serotonin 2C receptor regulates food uptake, and its activity is regulated by alternative pre-mRNA splicing. Alternative exon skipping is predicted to generate a truncated receptor protein isoform, whose existence was confirmed with a new antiserum. The truncated receptor sequesters the full-length receptor in intracellular membranes. We developed an oligonucleotide that promotes exon inclusion, which increases the ratio of the full-length to truncated receptor protein. Decreasing the amount of truncated receptor results in the accumulation of full-length, constitutively active receptor at the cell surface. After injection into the third ventricle of mice, the oligonucleotide accumulates in the arcuate nucleus, where it changes alternative splicing of the serotonin 2C receptor and increases pro-opiomelanocortin expression. Oligonucleotide injection reduced food intake in both wild-type and ob/ob mice. Unexpectedly, the oligonucleotide crossed the blood–brain barrier and its systemic delivery reduced food intake in wild-type mice. The physiological effect of the oligonucleotide suggests that a truncated splice variant regulates the activity of the serotonin 2C receptor, indicating that therapies aimed to change pre-mRNA processing could be useful to treat hyperphagia, characteristic for disorders like Prader–Willi syndrome.

**Keywords** alternative splicing; brain function; food uptake; obesity; pre-mRNA processing
**Subject Categories** Genetics, Gene Therapy & Genetic Disease; Metabolism; Pharmacology & Drug Discovery

## Introduction

Obesity is a growing health problem causing an estimated 100–400,000 deaths per year in the United States. Currently, more than two-thirds of the adult population in the USA is either overweight or obese (Ogden *et al*, 2014). The FDA approved drug locaserin is a serotonin 2C receptor (5HT2C) agonist, validating the 5HT2C as an anti-obesity drug target (Miller, 2005).

Various hormonal systems signal nutritional status to the brain that ultimately regulates feeding behavior. An important brain area integrating hormonal signals is the arcuate nucleus, a structure located in the mediobasal hypothalamus. The arcuate nucleus contains about 1,000 POMC/CART (pro-opiomelanocortin/cocaine- and amphetamine-regulated transcript) neurons that express insulin, leptin, and serotonin receptors, allowing POMC/CART neurons to sense nutritional status (McNay *et al*, 2012). Activation of these receptors results in a reduction of food intake due to POMC/CART neurons stimulating anorexigenic neurons in the paraventricular nucleus (PVN) and inhibiting orexigenic neurons in the lateral hypothalamic area (Cone, 2005).

Central to food control in the arcuate nucleus is the 2C-subtype of serotonin receptor (5HT2C) (Iwamoto *et al*, 2009). After stimulation with serotonin or other synthetic agonists, the receptor activates phospholipase C and induces expression of the POMC gene. This leads to an increase in expression of the POMC precursor peptide that is further processed to α-MSH (melanocyte-stimulating hormone), which causes an anorexic response by activating neurons in the paraventricular nucleus via the melanocortin 4 receptor (MC4R) (Williams & Elmquist, 2012; Berglund *et al*, 2014).

The pre-mRNA that generates 5HT2C undergoes extensive processing that includes alternative splicing of exon Vb and adenosine to inosine editing at five sites (Stam *et al*, 1994). The 5HT2C gene has one promoter but contains two polyadenylation sites (Zhang *et al*, 2013). The longest transcript contains six exons and undergoes both pre-mRNA editing and splicing affecting exon Vb. Exon V forms an extended secondary structure *in vivo* allowing RNA editing by adenosine deaminases acting on RNA (ADAR1 and ADAR2) that require an RNA duplex as a substrate. The RNA secondary structure undergoes conformational changes after binding to synthetic ligands, which regulates the use of the distal alternative splice site (Shen *et al*, 2013).

RNA editing creates at least 24 full-length proteins by affecting three amino acids located in the second intracellular loop of the

1  Department of Molecular and Cellular Biochemistry, University of Kentucky, Lexington, KY, USA
2  Department of Pharmacology, Vanderbilt University, Nashville, TN, USA
3  Department of Molecular Physiology & Biophysics, Vanderbilt University, Nashville, TN, USA
4  Spinal Cord & Brain Injury, Research Center, University of Kentucky, Lexington, KY, USA
   *Corresponding author. Tel: +1859 323 0896; E-mail: stefan@stamms-lab.net
   †These authors contributed equally to this work

receptor that couples to the effector Gq protein. The non-edited receptor features the amino acids I, N, and I (isoleucine, asparagine, and isoleucine) at positions 156, 158 and 160 of the receptor protein. The INI isoform shows the greatest coupling efficacy to the G protein in response to serotonin and is constitutively active (Labasque *et al*, 2010) whereas the highly edited isoforms show a reduced response to serotonin binding and little or no constitutive activation (Wang *et al*, 2000; Werry *et al*, 2008). Sole expression of the fully-edited isoform of the receptor causes hyperphagia in mice, demonstrating that altered 5HT2C signaling is critical for maintaining appropriate energy balance (Morabito *et al*, 2010).

Skipping of exon Vb creates a mRNA predicted to encode a shorter protein isoform containing only three transmembrane domains. Tagged proteins encoded by cDNAs for the shorter protein localize to the endoplasmic reticulum (ER). These proteins can dimerize with the full-length serotonin receptor, which likely inhibits serotonin signaling as a result of ER retention (Martin *et al*, 2013).

The importance of proper hypothalamic food regulation becomes apparent in Prader–Willi syndrome (PWS), a frequent syndromic form of obesity, characterized by hyperphagia (Butler *et al*, 2006; Cassidy *et al*, 2012). The syndrome is caused by the loss of gene expression from a maternally imprinted region on chromosome 15q11.2. This region contains several proteins and two clusters of C/D box small nucleolar RNAs (snoRNA). Recent genetic studies showed that the loss of these snoRNAs strongly contributes to PWS (Sahoo *et al*, 2008; de Smith *et al*, 2009; Duker *et al*, 2010). One of these snoRNAs, SNORD115, regulates the processing of the 5HT2C pre-mRNA (Kishore & Stamm, 2006; Kishore *et al*, 2010; Shen *et al*, 2011), showing functions of C/D box snoRNAs different from the 2′-O-RNA methylation normally attributed to these non-coding RNAs (Falaleeva *et al*, 2016), and suggesting that a dysregulation of 5HT2C function contributes to the hyperphagia observed in PWS.

Since pre-mRNA processing of the 5HT2C has a strong effect on food intake, we searched for molecules that could compensate for the loss of SNORD115. We identified an RNA oligonucleotide that similar to SNORD115 promotes exon Vb inclusion. This oligonucleotide reduces food intake in mice when delivered to the arcuate nucleus, either through ICV or systemic injections, most likely by increasing full-length 5HT2C density at the cell surface of neurons in the arcuate nucleus.

# Results

## The 5HT2C gene expresses both a full-length and a truncated protein isoform

The 5HT2C pre-mRNA contains two alternative 5′ splice sites termed the proximal (PS) and distal (DS) splice site that define exons Va and Vb, respectively. Inclusion of exon Vb generates RNA2, which encodes a full-length seven-transmembrane receptor. Skipping of exon Vb introduces a frameshift, resulting in an early stop codon in exon VI. The resulting RNA1 thus encodes a truncated receptor isoform containing only three predicted transmembrane domains (Fig 1A). Whereas the full-length 5HT2C receptor is 90% identical between mouse and human and differs by only one amino acid in length, the C-terminus of the truncated receptor contains 96 amino acids in humans, but only 19 amino acids in mouse (Fig EV1A) due to differential codon usage. Since the stop codon generating the truncated receptor is located in the last exon, its mRNA is likely not subject to nonsense-mediated decay (Fatscher *et al*, 2015). To test whether RNA1 is expressed as an endogenous protein, we generated an antiserum against a peptide common to mouse and human RNA1 (Fig EV1B). After affinity purification, this antiserum detected truncated 5HT2C protein from HEK293 cells transfected with a cDNA encoding the truncated receptor, as well as the endogenous protein from mouse brain (Fig 1B), which shows the existence of a truncated 5HT2C *in vivo*.

## Identification of oligonucleotides that change exon Vb inclusion

To study alternative pre-mRNA splicing of serotonin 2C receptor exon Vb, we created a series of minigene constructs that contain the alternative exon V. To test the effect of oligonucleotides on this construct, we omitted a start codon, which limits interference from protein-coding requirements. The minigene contained exon IV, intron IV, exon Va, exon VI, and intron V that was shorted from about 59 to 6 kb, (Fig 1C) (Kishore & Stamm, 2006). We performed an oligonucleotide walk (Singh *et al*, 2012) to identify oligonucleotides that promote exon Vb inclusion. We used adjacent 18-mers showing complementarity against exon Vb and intron V (Fig 1C). The oligonucleotides were 2′-O-methyl phosphothioates, where each ribose and each phosphate group was modified by a 2′-O-methyl modification or a single sulfur, respectively. Previous studies using oligonucleotides with this chemistry found that cells take up the oligonucleotide directly without adjuvants (Heemskerk *et al*, 2009) and related 2′-O-2-methoxyethyl oligonucleotides are taken up by neurons in the brain (Passini *et al*, 2011), which we also observed (Fig EV2). As shown in Fig 1D, oligo#5: 5′-AGUAUU-GAGCAUAGCCGC-3′ strongly activates exon Vb usage, as determined by RT–PCR. Thus, we identified an oligonucleotide that enters cells and strongly promotes exon Vb inclusion, similar to SNORD115.

## Oligo#5 acts at nanomolar concentration in cell culture and can be modified at its 5′ end

We next determined the working concentration of oligo#5 by testing a concentration range from 5 to 100 nM, using the reporter minigene transfected into HEK293 cells. Exon Vb inclusion starts increasing at concentrations higher than 10 nM (Fig 2A and B), showing that oligo#5 can exert an effect in the low nM concentration range; the $EC_{50}$ (half maximal effect) was 13 nM.

We then shortened oligo#5 from 18 nt to 12 nt and 10 nt, removing 6 or 8 nt from the 5′ end, respectively (oligo#5-3; oligo#5-10, Fig 1C). The shorter oligo#5-3 retained some activity, which was abolished in oligo#5-10 (Fig 2C and D). Next, we explored possible modifications and tested oligo#5 derivatives that were modified at the 5′ and 3′ end by cyanine 3. The cyanine dyes were attached to the phosphate via an ester moiety. A modification at the 5′ end had no effect on activity, whereas modification on the 3′ end abolished activity (Fig 2E and F). The data show that oligo#5 works in a low nM concentration and can be modified at its 5′ end.

   

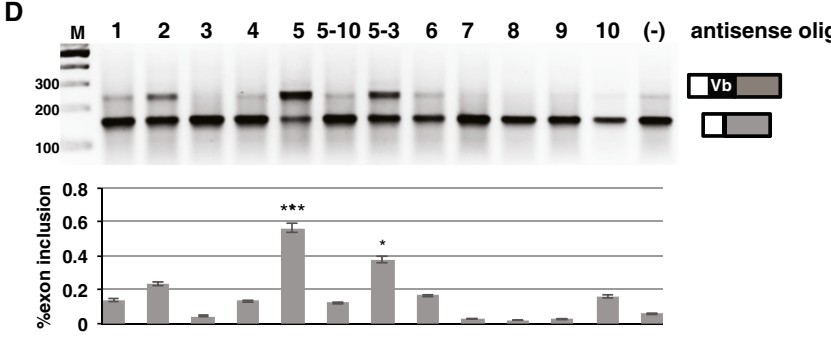

**Figure 1.   Identification of oligonucleotides that change alternative splicing of the serotonin 2C receptor.**

A   Pre-mRNA processing of the serotonin 2C receptor. Exons are shown as boxes, and introns are shown as lines. Due to the use of two alternative splice sites (PS, DS: proximal and distal splice site), exon Vb is alternatively used, resulting in two RNAs: RNA1 lacking and RNA2 including exon Vb. Alternative splicing patterns are indicated by dashed and dotted lines; constitutive splicing patterns are indicated by solid lines. The start codon is indicated using a line with circle. Alternative usage of exon Vb introduces a frameshift, leading to different stop codons being used (lines with rectangles in exon VI and red box in exon VI indicate a different reading frame). The proteins formed by RNA1 (truncated 5HT2C) and RNA2 (full-length 5HT2C) are indicated. Circles in the second intracellular loop of RNA2 protein depict three amino acids changed by RNA editing. The location of the epitope against the truncated 5HT2C is indicated with a Y.

B   The truncated receptor encoded by RNA1 is endogenously expressed. HEK293 cells were transfected with a cDNA encoding the human truncated 5HT2C (RNA1). SDS lysates from these transfected cells and from mouse hypothalamus were analyzed by Western blotting using an antiserum generated against RNA1 (Fig EV1). Mouse hypothalamus was exposed 10 times longer. In brain, a non-specific band around 75 kDa is detected by the antiserum (indicated by a star).

C   The minigene used for the oligo walk is schematically shown; the arrow indicates the CMV promoter. The boxed area corresponds to the sequence is shown below. The secondary structure of the serotonin 2C receptor pre-mRNA determined by SHAPE (Shen et al, 2013) is shown, and the oligonucleotides used are indicated as lines. The binding site for SNORD115 (Kishore & Stamm, 2006) is boxed, and the five adenosines that undergo RNA editing are highlighted. The exon/alternative exon border in the proximal splice site (GG) and the alternative exon/intron border (Gg, distal splice site) are underlined and marked in red. Dots indicate base pairing, including non-canonical G-U base pairs.

D   RT–PCR analysis of the serotonin 2C receptor minigene after oligonucleotide addition. Below is a quantification of three independent experiments. A statistical evaluation showed significant differences between oligo#5 (***$P$ = 0.00008) and oligo#5-3 (*$P$ = 0.0019) and control (Student's $t$-test).

Source data are available online for this figure.

## Oligo#5 promotes exon Vb inclusion without RNA editing

Exon Vb undergoes RNA editing that likely changes the receptor pre-mRNA structure, as it weakens the central double-stranded RNA stem by replacing A:U interactions with thermodynamically less stable I:U interactions (Serra *et al*, 2004). Mutagenesis studies that changed the edited adenosines into guanosine residues showed that editing activated exon Vb inclusion (Flomen *et al*, 2004). We therefore tested the effect of oligo#5 on RNA editing by directly sequencing the RT–PCR product generated from transfecting the 5HT2C reporter gene with oligo#5. As shown in Fig 3A, none of the editing sites showed a change from adenosine to guanosine, indicating the absence of pre-mRNA editing. Similarly, editing was absent in exon Vb when we analyzed the minor exon Vb containing PCR product which is generated in the absence of oligo#5 (Fig 3B). Thus, the protein product generated by oligo#5 action encodes the non-edited receptor that shows the strongest response to serotonin, as well as the greatest constitutive activity.

## Oligo#5 acts directly on the 5HT2C pre-mRNA

Alternative pre-mRNA splicing is linked to other steps of gene expression, such as RNA polymerase speed and chromatin structure (de Almeida & Carmo-Fonseca, 2012). Thus, oligo#5 could promote exon Vb inclusion indirectly. To test how oligo#5 interacts with the pre-mRNA, we constructed an *in vitro* splicing minigene that contains exons Va, Vb, and 115 nt of the downstream intron, fused to the adenovirus major late-transcription unit-derived MINX exon that contains a strong splicing acceptor (Zillmann *et al*, 1988). This construct differs from the endogenous gene by the lack of upstream sequences. Most importantly, the length of the downstream intron that is at least 57,969 nt in the endogenous gene was shortened to 190 nt (Fig 3C). Furthermore, we included the strong MINX acceptor splice site to obtain splicing *in vitro*. Uniformly $^{32}$P-labeled, 7-methyl guanosine-capped RNA made from this construct was incubated under standard splicing conditions in HeLa nuclear extract. We then tested the effect of oligonucleotides in Fig 1C on the processing *in vitro*. In the absence of any oligonucleotide, the proximal splice site is exclusively used, resulting in a 151 nt long fragment. This is similar to transfection experiments, where exon Vb skipping is the default mode. Addition of all oligonucleotides to a final concentration of 25 nM resulted in the release of exon Va (105 nt) and formation of a lariat containing exon Vb (L1). However, the formation of the exon Va-MINX product (151nt) was suppressed. This indicates that the oligonucleotides promote cleavage at the proximal splice site as the first step of splicing, but inhibit the second step that would result in exon ligation.

Oligo#5, #5-10 and 5-3, and #6, 7, 8, 9 also promote cleavage at the distal splice site, allowing for the first step in splicing that leads to exon Vb inclusion (201 nt). Importantly, similar to the cell-based experiments, oligo#5-3 promotes formation of the exon Vb-MINX product (247 nt) that reflects RNA2 formation. The quantification of four experiments shows the doubling of the exon Vb-MINX splice product, when the inactive oligo#3 is compared with oligo#5-3 (Fig 3E). The presence of these oligonucleotides also resulted in an accumulation of a lariat LII that lacks exon Vb. It is possible that the oligonucleotides stabilize this structure, which will favor exon Vb inclusion, leading to the formation of the correct spliced product

Va–Vb-MINX. However, oligo#5-3 had the strongest effect *in vitro*, whereas oligo#5 had only a small effect, contrasting the findings in cell culture. Since the *in vitro* reactions are performed at 30°C, it is possible that oligo#5 binds too tightly to the substrate RNA, which inhibits the second step of splicing.

Together, the data suggest that oligo#5 promotes cleavage at the distal splice sites, while still allowing ligation to the downstream exon in the second step of splicing, possibly by binding to the lariat LI.

## Oligo#5 changes cellular localization of the 5HT2C protein

To determine the effect of oligo#5 on 5HT2C protein, we generated a reporter construct that expresses a full-length receptor–GFP fusion protein when exon Vb is included. The construct (Fig 4A) contains the human serotonin 2C receptor protein-coding cDNA and a shortened intron V between exon Vb and VI containing the natural splice sites. We removed the 5HT2C stop codon and fused it in-frame with GFP. The start codon is located in exon III in its natural sequence context. Similar to the wild-type system, inclusion of the alternative exon Vb creates a protein with seven-transmembrane domains that are now linked to GFP. The GFP domain is predicted to be on the inside of the cell, joined to the last transmembrane domain. Skipping of exon Vb creates a frameshift resulting in a truncated protein without GFP (Fig 4A). Thus, the system can detect the effect of alternative exon usage on a GFP reporter protein. The effect of oligo#5 on the protein-expressing construct was first tested by RT–PCR. There is a higher basal inclusion level of exon Vb without oligo#5, which is likely the result of the upstream exonic sequences in this minigene. We observed strong inclusion of exon Vb after oligo#5 addition (Fig 4B).

For further protein studies, we generated stable HeLa cell lines expressing this construct after integration into a single specific flipase recognition target (FRT) site at a transcriptionally active genomic locus. Previous studies showed that the truncated 5HT2C receptor localizes to the endoplasmic reticulum, whereas the full-length receptor is found both intracellularly and at the surface (Martin *et al*, 2013). We therefore fractionated cells into a membrane-enriched and cytosolic fraction (Mancia *et al*, 2008) and detected the truncated 5HT2C using our antiserum and the full-length 5HT2C using a polyclonal anti-GFP antiserum. The truncated receptor is found both in the membrane-containing pellet and cytosolic fractions. Oligo#5 decreases the amount of the truncated 5HT2C receptor protein found in cytosolic fractions (Fig 4C and E). The full-length receptor is found only in the membrane associated fraction, and its expression is increased about 2.5-fold after oligo#5 treatment, reflecting the change in splice site selection (Fig 4D and E). The 5HT2C–GFP fusion protein migrates as several bands on Western blot, which likely reflects differential states of glycosylation (Backstrom *et al*, 1995). The data indicate that changes at the RNA level are reflected at the splicing level. Furthermore, a fraction of the truncated receptor appears to be only loosely associated with membranes, as it is found in the membrane-free supernatant, which could be due to the lack of four transmembrane domains.

Next, we analyzed the effect of oligo#5 on the localization of the full-length receptor using stable cell lines. Transfection of 5 nM oligo#5 resulted in an accumulation of GFP signal on the cell surface, whereas the control oligo had no effect (Fig 4F and G). The

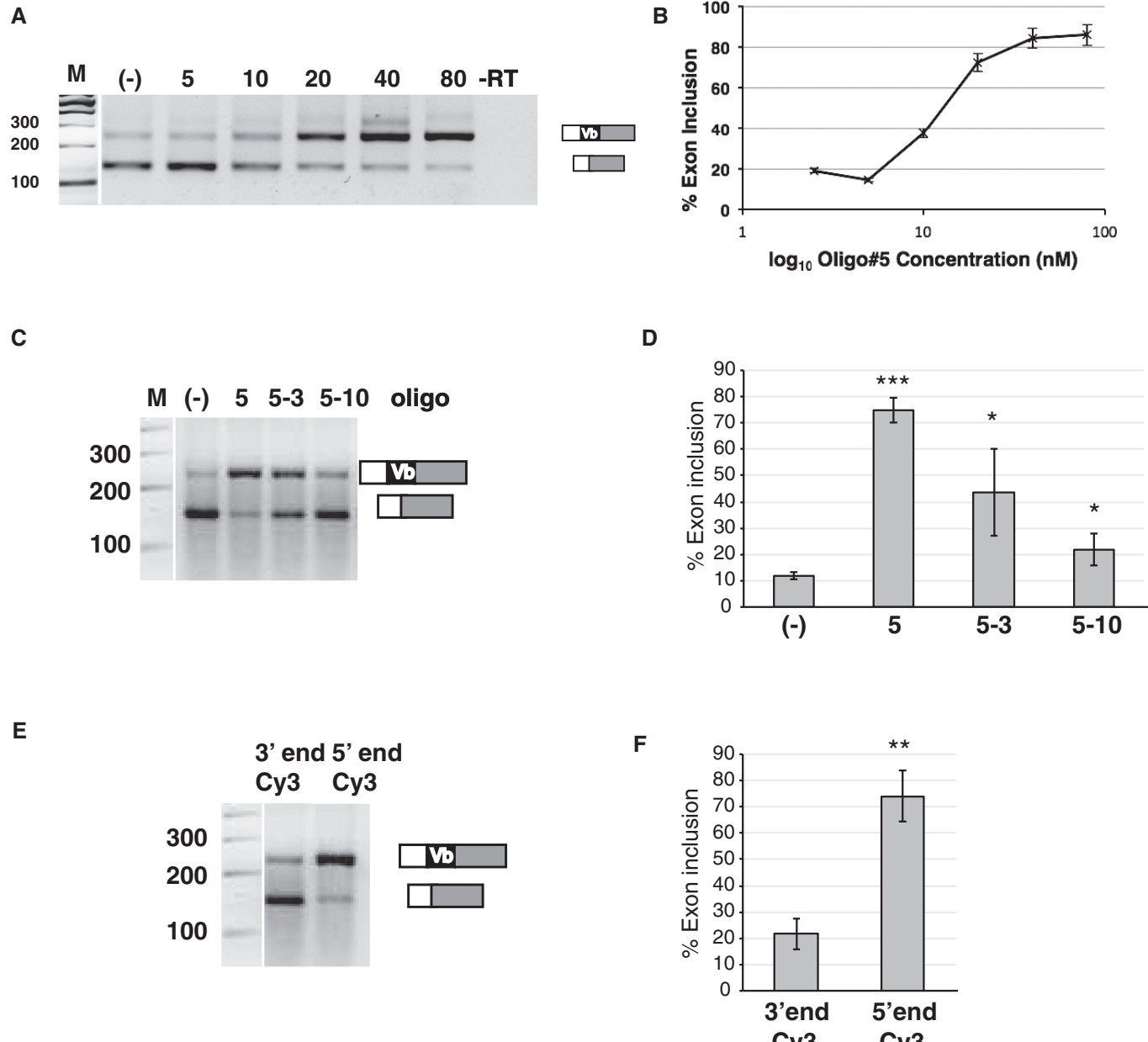

**Figure 2. Characterization of oligo#5 and its derivatives.**

A   Titration of oligo#5. An increasing amount of oligo#5 was added to HEK293 cells stably expressing the serotonin 2C receptor reporter minigene, and the RNA was analyzed by RT–PCR. –RT: negative control without reverse transcriptase.

B   Quantification of three independent experiments. The half-maximal effect was seen at 13.1 nM concentration. Error bars indicate standard deviation.

C   Influence of oligo#5 shortening on alternative splicing of the serotonin 2C receptor. Oligo#5: 5′ AGU AUU GAG CAU AGC CGC 3′ (18mer); Oligo#5-3: 5′ GAG CAU AGC CGC 3′ (12mer); and Oligo#5-10: 5′ G CAU AGC CGC 3′ (10mer) were used in transfection assays analyzed by RT–PCR.

D   Statistical analysis of the effect of oligonucleotide shortening. The effect of all oligos testes is significant, oligo#5: ***$P$ = 0.0015; oligo#5-3: *$P$ = 0.031; and oligo#5-10: *$P$ = 0.048 (Student's $t$-test), $n$ = 6. Error bars indicate standard deviation.

E   Influence of cy3 label on the 5′ and 3′ end of oligo#5. Using a phosphate linker, Cy3 was added to the 5′ or 3′ phosphate, respectively, and the oligonucleotides were assayed by RT–PCR. Oligo#5 with cy3 at the 5′ end showed 86% inclusion, with Cy3 at the 3′ end 17% exon inclusion.

F   Statistical analysis of the effect of oligonucleotide modification. The effect is significant, **$P$ = 0.009 (Student's $t$-test), $n$ = 4. Error bars indicate standard deviation.

Source data are available online for this figure.

GFP signal on the cell surface rose from about 35 to 55% in the presence of oligo#5 (Fig 4H). Western blotting analysis showed a relative increase of the protein ratios of full-length to truncated receptor (Fig 4C–E). Thus, in agreement with earlier findings using cDNAs (Martin *et al*, 2013), the data show that oligo#5 causes an accumulation of the full-length receptor on the cell surface likely by

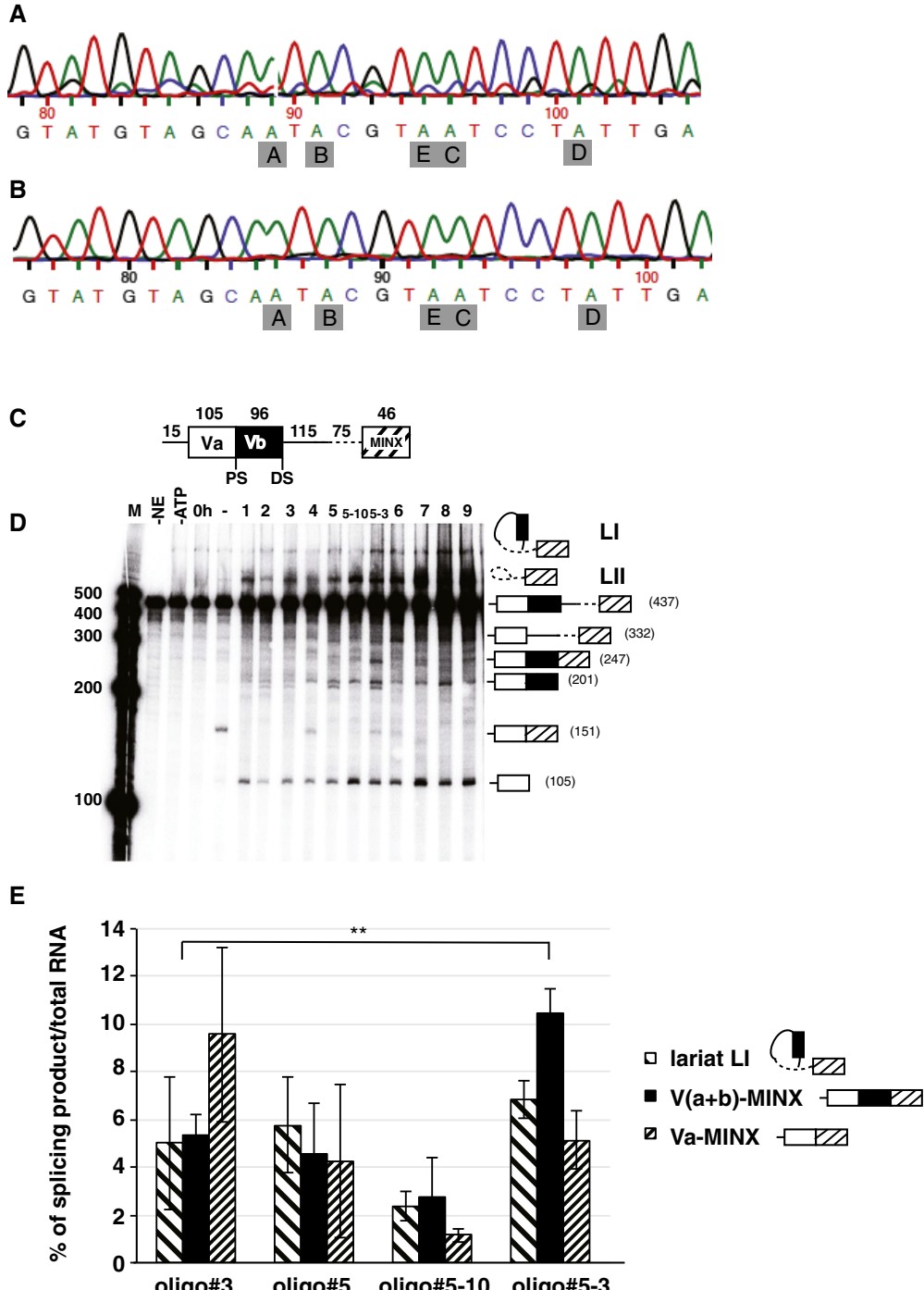

**Figure 3.   Oligo#5 does not influence RNA editing of the serotonin 2C receptor and acts directly on the pre-mRNA.**

A    The RT–PCR product generated from cells treated with oligo#5 was directly sequenced using the Sanger sequencing method. The fluorescent peaks after capillary electrophoresis are shown. Letters in gray boxes indicate the editing sites (also shown in Fig 1C).

B    Sequence of the weak control band, generated without oligo#5 (Fig 1C, lane (-)).

C    Structure of the *in vitro* minigene: The construct is schematically shown on top and consists of exons Va, Vb, and 115 nt of the downstream intron and a MINX exon with 75 nt (dashed line) of the upstream intron. A striped box indicates the MINX exon.

D    *In vitro* splicing assay: Uniformly labeled *in vitro* transcribed RNA was incubated with HeLa nuclear extract (NE) under splicing conditions for 2 h at 30°C. The structures of the *in vitro* splicing products are shown on the right. LI and LII are two lariats formed. Numbers indicate the expected product lengths.

E    Quantification of selected bands from four independent experiments. The percentage of each form in the total products is shown. The differences of splice products (Va+b-MINX) is significant, **$P = 0.0055$ (Student's *t*-test); $n = 3$. Error bars indicate standard deviation.

Source data are available online for this figure.

reducing the amount of the truncated receptor isoform that sequesters the receptor intracellularly.

## Oligo#5 enters neurons after intracerebroventricular injection

To test the effect of oligo#5 *in vivo*, we first determined whether the oligonucleotide is taken up by neurons in mouse brain. To circumvent the blood–brain barrier, we performed intracerebroventricular (ICV) injections of oligo#5. We injected 2 µg oligo#5 5′ end-labeled with Cy3 into the 3$^{rd}$ ventricle of male C57BL/6 mice (Fig EV3). Oligo#5 uptake was determined by confocal microscopy in coronal brain sections. We observed accumulation of oligo#5 in areas adjacent of the 3$^{rd}$ ventricle (Fig 5A), including the arcuate nucleus (Fig 5B). Similar to other oligonucleotides with a 2′-O-methyl-phosphothioate chemistry, oligo#5 was taken up by the cells without adjuvant, which we also observed in cell culture (Fig EV2). Immunohistochemisty with the neuronal marker NeuN showed that the oligonucleotide was detectable in neurons, as well as NeuN-negative cells, which are likely to be glia (Fig 5C). To determine the uptake of oligo#5 in glia we stained brain sections with GFAP (glial fibrillary acidic protein) after oligo#5 injection and detected oligo#5 staining in some, but not all glial cells (Fig 5D). It is thus likely that oligo#5 enters most cells in the CNS after injection.

Our transfection data indicate that oligo#5 changes the alternative splicing of the 5HT2C (Figs 1C and 4B), and we therefore tested its effect on the endogenous 5HT2C after injection. We microdissected the area in the ventral part of the 3$^{rd}$ ventricle that contained the arcuate nucleus (Fig 5A and B), and performed qRT–PCR on the isolated RNA. Using primers against the 5HT2C, we determined the expression of RNA1 and RNA2 and observed a strong increase of RNA2 after 3 and 6 h (Fig 5E). In contrast, there was no effect on the serotonin 4 receptor (Htr4), showing the selectivity of oligo#5 (Fig EV4). In all injection experiments, we used an oligonucleotide against human SMN2 as a control (Seo *et al*, 2014). The SMN2-20mer oligonucleotide had the same 2′-O-methyl monophosphothioate chemistry as oligo#5. Oligonucleotides with this chemistry are taken up by cell without adjuvants (Heemskerk *et al*, 2009), and SMN2 oligonucleotides with the related 2′-O-(2-methoxyethyl) phosphorothioate chemistry were previously shown to enter spinal cord neurons after epidural injection in mice (Hua *et al*, 2010). SMN2 is not expressed in mice, and thus, the SMN2 oligonucleotide has no specific target and controls for unspecific effects of injection.

To analyze the editing patterns of 5HT2C transcripts, we analyzed cDNA corresponding to exon Vb inclusion using deep sequencing (Fig 5F). Similar to the situation in cell culture, we did not observe a change in editing of 5HT2C RNAs (Fig 3A and B), suggesting that oligo#5 acts mainly on the RNA2/RNA1 ratio (Fig 5E and F).

These data suggest that oligo#5 changes alternative splicing of the endogenous 5HT2C and promotes inclusion of exon Vb, as expected from the *in vitro* experiments. These changes increase the amount of non-edited 5HT2C, which encodes a serotonin receptor that is active without ligand, both in cell culture (Wang *et al*, 2000; Werry *et al*, 2008; Labasque *et al*, 2010) and *in vivo* in spinal cord neurons (Murray *et al*, 2011). Activation of the serotonin receptor in the arcuate nucleus results in an increase of POMC expression (Valassi *et al*, 2008; Xu *et al*, 2008). We therefore measured POMC RNA levels in the ventral part of the 3$^{rd}$ ventricle and found a significant increase in POMC RNA expression after three to 6 h post-injection (Fig 5G).

## Oligo#5 decreases food intake in mice

An increase in POMC expression is expected to decrease food intake through the activation of MC4R receptors (Cone, 2005). We thus investigated the effect of oligo#5 on food consumption by measuring total food intake (Fig 5H). We used C57BL/6 mice that had no access to food for 16 h as a model for hyperphagia. After surgical implantation of a guide cannula and a recovery period, the mice had no access to food for 16 h. Then, we injected the oligonucleotide with the mice having free access to food, and their food consumption was measured. Oligo#5 reduced food intake by about 70% when compared to animals that just received the cannula surgery or both the cannula surgery and a control oligonucleotide against human SMN2 (Fig 5H).

To account for possible changes resulting from food withdrawal, we investigated ob/ob mice as a genetic model for hyperphagia. Ob/ob mice lack leptin and are hyperphagic (Zhang *et al*, 1994). Throughout the experiments, the mice had free access to food. Similar to fasted, wild-type mice, oligo#5 injection reduced food uptake by about 75% in these mice (Fig 5I).

---

**Figure 4. Oligo#5 influences the ratio of the two serotonin 2C receptor isoforms and changes the localization of the full-length receptor.**

A    Structure of the reporter gene introduced into cells. The start codon in exon III is shown (circle) and the splicing patterns emerging from using the distal and proximal splice site (DS, PS). Skipping of exon Vb causes a frameshift, which terminates the protein in the first stop codon (square). Exon Vb inclusion keeps a longer open reading frame that terminates after the GFP sequence. The encoded proteins are schematically shown below.

B    Effect of oligo#5 on pre-mRNA splicing of the reporter gene, stably transfected into HeLa cells.

C    Effect of oligo#5 on the truncated 5HT2C protein. Stable cell lines expressing the construct shown in panel (A) were treated with oligo#3 and oligo#5. After 48 h, cells were separated in membrane-containing fractions and soluble cytosolic supernatant. The fractions were analyzed by Western blot using the anti-RNA1 antiserum (Fig EV1). The protein samples were on the same membrane, but the supernatant fractions were exposed about ten times longer.

D    Effect of oligo#5 on the full-length 5HT2C protein. The protein was prepared as in panel (C) and analyzed using a polyclonal anti-GFP antiserum.

E    Quantification of panels (C) and (D) from three independent experiments. The ratio of protein signal after oligo#3 and oligo#5 treatment is shown for RNA1 (*P = 0.002) and RNA2 (***P = 0.0003) (Student's *t*-test), n = 3, respectively. Error bars indicate standard deviation.

F, G    Effect of oligo#5 on localization of the EGFP-tagged serotonin 2C receptor. Confocal images of three representative pictures are shown.

H    Quantification of GFP signal found on the membrane after oligo#5 addition. The signal around the peripheral region was determined by visual inspection and calculated as a fraction of total signal using NIH image. Two hundred cells from three individual experiments were analyzed, **P < 0.008 (Student's *t*-test), n = 200. Error bars indicate standard deviation.

Source data are available online for this figure.

To further test the involvement of the melanocortin system in oligo#5 action, we injected oligo#5 into MC4R receptor knockout mice (Srisai *et al*, 2011). POMC is processed into α-MSH, which acts on the MC4R receptor, leading to an anorexic response. Using these mice, we did not see a significant reduction in food intake (Fig EV5), further suggesting that oligo#5 works by activating POMC neurons (Fig 5G).

In summary, oligo#5 reduces food intake in fasted (i.e. hungry) mice and in genetically altered, hyperphagic mice and likely acts by stimulating POMC expression.

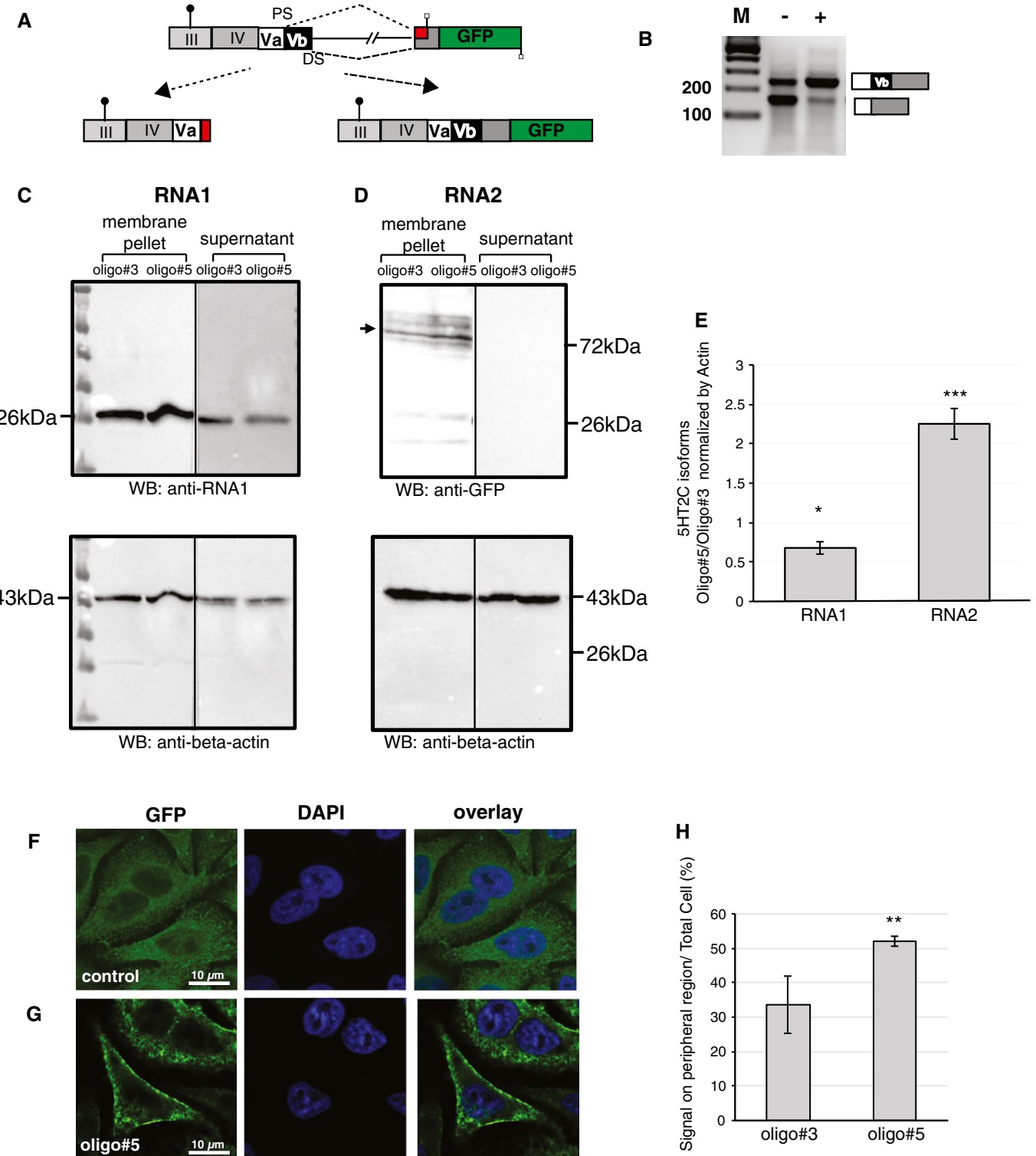

**Figure 4.**

## Oligo#5 reduces food intake after systemic injection

We next tested the effect of oligo#5 after systemic delivery into the blood. Previous studies showed that, in general, oligonucleotides do not cross the blood–brain barrier (Banks, 2009). However, the highest expression of the 5HT2C is in the choroid plexus (Esterle & Sanders-Bush, 1992), the structure that generates cerebrospinal fluid from blood. In the choroid plexus, almost exclusively RNA1 is expressed (Kishore & Stamm, 2006). To bypass possible liver clearance, we injected the Cy3-tagged oligonucleotide via a carotid artery catheter directed toward the brain. Unexpectedly, we saw an accumulation of the oligonucleotide in the brain after 1 h of injection (Fig 6A). As expected, the strongest staining was in the choroid plexus that has the greatest exposure to the systemic circulation and showed the highest 5HT2C expression. In addition, we saw staining in the epithelium surrounding the ventricles and unexpectedly in the retrosplenial granular cortex. Delivery through carotid injection results in a more widespread distribution of oligo#5 in the brain than ICV injection into the third ventricle. This is likely due to the 50- to 100-fold higher amount of oligo#5 injected and could reflect oligo entry through each of the four choroid plexuses. To determine a possible effect on 5HT2C processing, we microdissected the region around the 3$^{rd}$ ventricle and analyzed 5HT2C splicing using real-time PCR (Fig 6B). Again, we found an increase of the RNA2/RNA1 ratio, suggesting that oligo#5 influences 5HT2C splicing after crossing the blood–brain barrier.

We next tested the effect of oligonucleotide injection on food uptake in metabolic cages using mice that had free access to food all the time. We found an about 80% reduction compared to a control SMN2 oligonucleotide after 12 h (Fig 6B). There was no difference in water uptake, the mobility and activity between mice that received control oligonucleotide or oligo#5 (Fig EV6). The effect of oligo#5 injection persisted over 3 days (Fig 6D). Together, the data show that oligo#5 reduces food uptake after systemic injection, likely after crossing the blood–brain barrier and changing serotonin receptor splicing in the brain.

## Discussion

The 5HT2C protein is a validated anti-obesity drug target that is exploited by the approved agonist locaserin (Miller, 2005). Two major isoforms of 5HT2C-mRNA are formed as a result of alternative splicing: RNA1 encoding a truncated receptor isoform and RNA2 containing the alternative exon Vb that encodes a full-length receptor (Fig 6E). RNA1 encodes a truncated receptor that is associated with intercellular membranes. This truncated receptor sequesters the full-length receptor inside the cell, which strongly decreases the serotonin response (Martin *et al*, 2013). The loss of a trans-acting splicing regulatory RNA, SNORD115, that promotes exon Vb inclusion contributes to Prader–Willi syndrome (PWS) (Kishore & Stamm, 2006), suggesting that a dysregulation of 5HT2C pre-mRNA processing could result in obesity. To identify therapeutic approaches that might substitute for SNORD115, we performed an oligo walk using RNA-based oligonucleotides containing 2′O-methyl and monophosphothioate modifications as they are RNAse resistant, are taken up by cells without adjuvants, and show, in contrast to DNA oligonucleotides, no RNAseH activity (Bennett & Swayze, 2010; Hua *et al*, 2010).

We identified oligo#5 located in the intron downstream of the alternatively spliced exon Vb. Oligo#5 promotes exon inclusion without RNA editing, suggesting that the pre-mRNA can undergo one of three choices: (i) either include exon Vb in an unedited form or (ii) edit exon Vb, which results in weakening an RNA structure that blocks the proximal splice site to produce an edited mRNA; or (iii) skip exon Vb. In the cell, all pathways are present and likely compete for the pre-mRNA. Oligo#5 favors the first pathway and increases exon Vb inclusion without editing to promote full-length receptor formation, but decreases the truncated 5HT2C isoform formation.

Skipping of exon Vb generates a truncated protein isoform that contains only three transmembrane regions. Using a novel antiserum directed against the truncated protein, we demonstrate that this receptor isoform is expressed in the brain, as predicted from its

Figure 5.    **Oligo#5 enters neurons after ICV injection, changes 5HT2C splicing and food intake.**

A    Accumulation of Cy3-labeled oligo#5 in the area around the 3$^{rd}$ ventricle 1 h post-injection is shown the ventral part of a coronal section −1.34 mm from Bregma. The injection coordinates were AP: −1.82 mm, DV: −4.5 mm, ML: 0.

B    Uptake of oligo#5 in the arcuate nucleus region (arc), enlarged from panel (A).

C    Oligo#5 enters neurons. Enlargement of hypothalamus region taking up oligo#5 (red), stained for the neuronal marker NeuN (green). Most of the oligo#5 signal is in neurons, but some is also in NeuN-negative cells (arrow), likely glia.

D    Oligo#5 enters some glia cells. Enlargement of a hypothalamic region adjacent to the 3$^{rd}$ ventricle 1 h after oligo#5 injection. The oligo (red) can be found in some GFAP-positive glia cells (arrow), but numerous GFAP-positive cells (green) show no oligo#5 uptake.

E    Effect of oligo#5 on RNA2/RNA1 ratios in the injected area. The area marked by Cy3 staining in the dorsal hypothalamus consisting mainly of the arcuate nucleus was microdissected, RNA isolated and analyzed by qPCR. NS: non-significant changes (3 h: *P = 0.01; 6 h: ***P = 0.000001; 9 h: *P = 0.02; 12 h: P = 0.06; n = 3).

F    Oligo#5 does not change the RNA2-editing pattern. The hypothalamic area showing oligo#5 staining was microdissected 6 h post-injection, and RNA2 was amplified and the PCR product subjected to deep sequencing. The percent of non-edited reads in total reads is shown (at least 20,000 reads per lane).

G    Effect of oligo#5 on POMC expression. The RNA from panel (D) was analyzed using qPCR with primers against POMC, normalized to GAPDH (3 h: ***P = 0.0009; 6 h: *P = 0.02; 9 h: P = 0.1; 12 h: P = 0.3, n = 3).

H    Effect of oligo#5 on food uptake in wild-type mice. C57BL/6 wild-type mice received a guide canulae and did not have access to food for 16 h. After injection of 2 μg oligonucleotides, they could freely access food. The food consumption was measured by weighing the food used. A total of 52 animals were used, P = 0.001 for the difference between oligo#5 and control oligonucleotides. Each of the six individual experiments showed statistically significant differences between oligo#5 and control (P < 0.01–0.001). Control oligo: an oligonucleotide against human SMN2.

I    Effect of oligo#5 on food uptake in ob/ob mice. Ob/ob mice received oligo#5 through ICV injection, similar to panel (F). However, the mice could freely access food at all times (P = 0.04, n = 3).

Data information: (E-I) Error bars indicate standard deviation. Student's *t*-test.

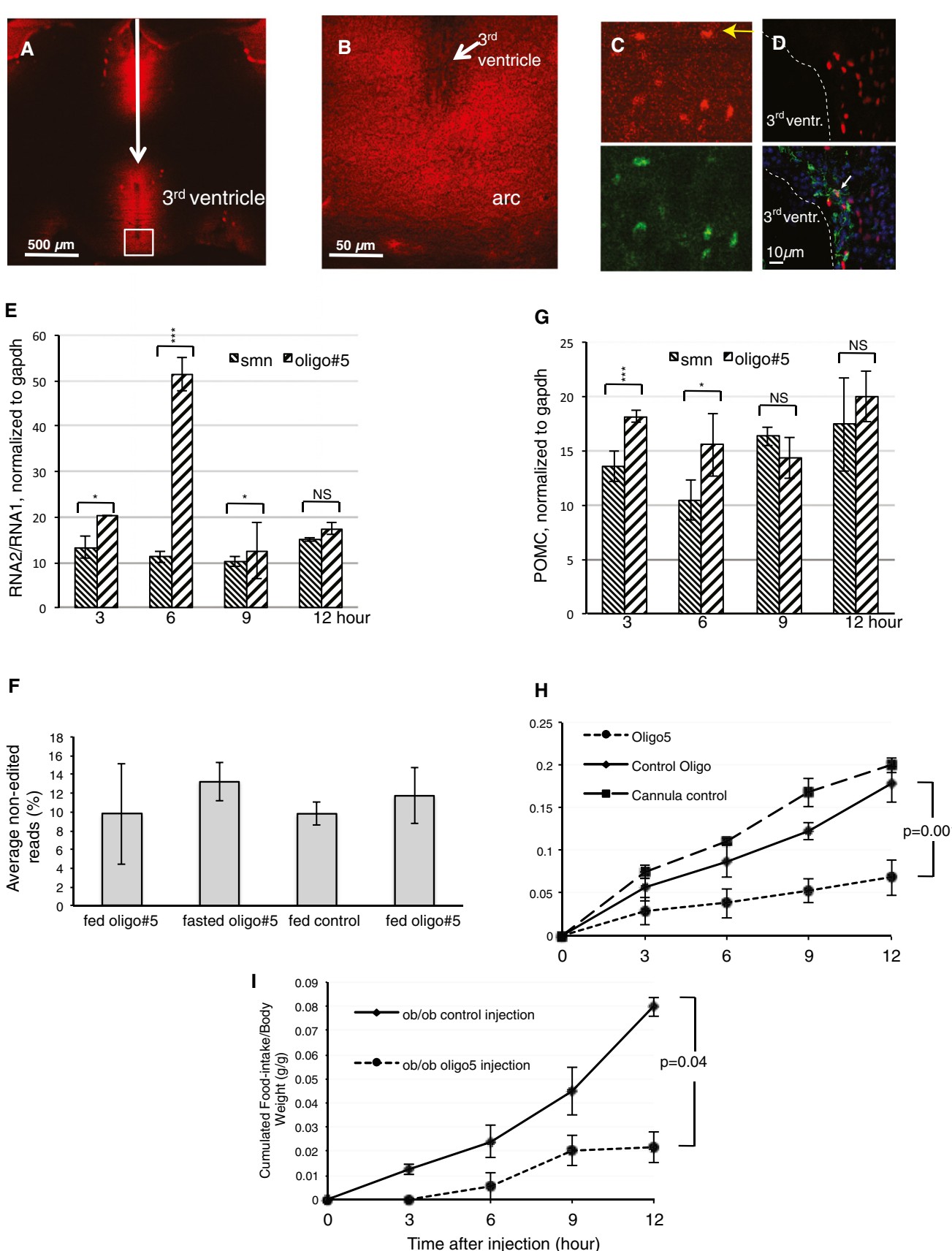

Figure 5.

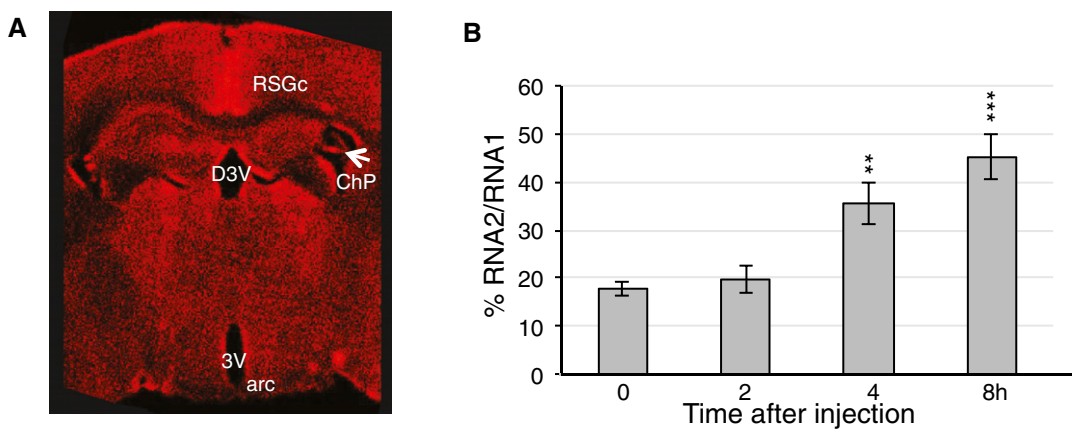

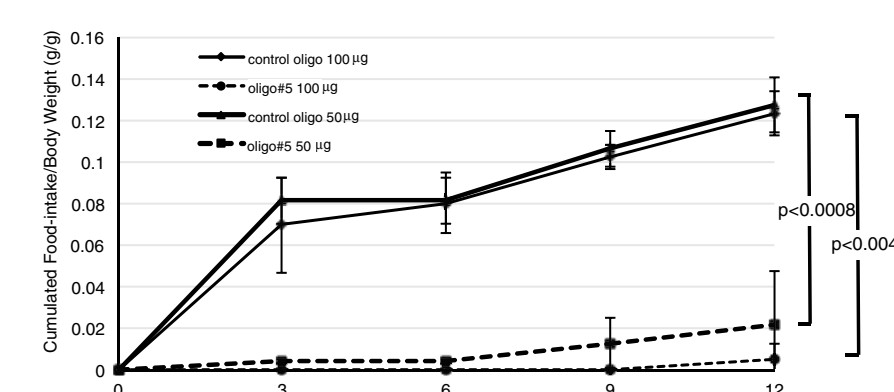

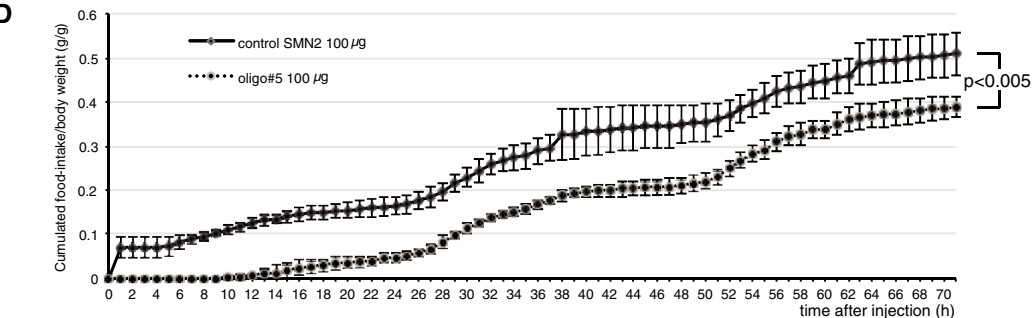

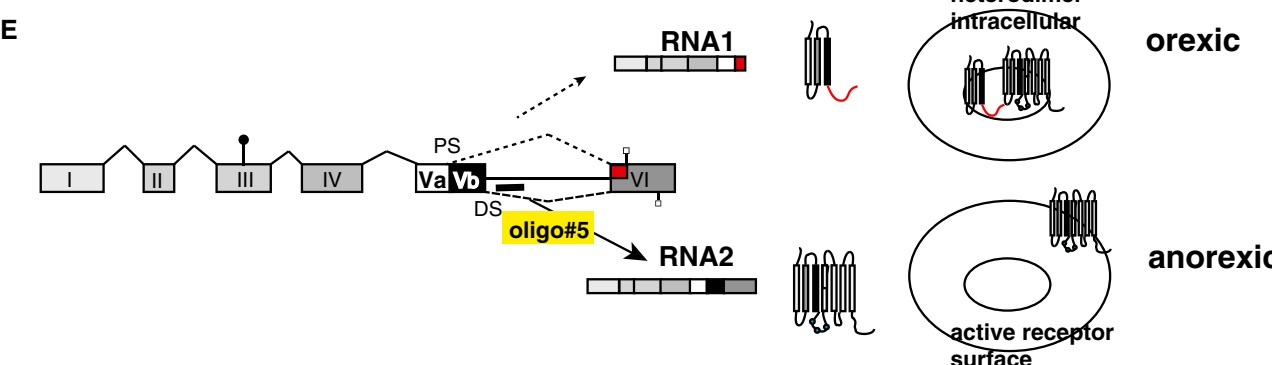

**Figure 6.**

◄

**Figure 6. Oligo#5 reduces food intake after delivery to the blood.**

A   About 50 µg of cy3-labeled oligo#5 was delivered through a carotid artery catheter, and 1 h later, the cy3 signal was detected. ChP: choroid plexus, RSGc: retrosplenial granular cx,c, 3V: 3rd ventricle; D3V: dorsal 3rd ventricle.

B   Change of RNA2/RNA1 ratios after carotid artery injection. The area around the arcuate nucleus was microdissected at the times indicated post-injection. The ration of RNA2/RNA1 was determined using real-time PCR. Changes became apparent statistically significant after 4 and 8 h (2 h: $P = 0.3$; 4 h: **$P = 0.003$, 8 h: ***$P = 0.0006$, $n = 5$).

C   Effect of oligo#5 delivered through a carotid artery catheter on food uptake in wild-type mice. C57BL/6 wild-type mice with a pre-implanted cranial dosing carotid catheter received 50 or 100 µg of the oligonucleotides indicated. $n = 3$.

D   Long-term effect of oligo#5 action. Food intake was measured for 70 h after one injection of 100 µg oligo#5 through a carotid catheter at time zero. $n = 3$.

E   Model of oligo#5 action. The 5HT2C receptor generates two major RNA isoforms generating a truncated receptor and a full-length receptor. The truncated isoform is retained in intracellular membranes. Due to heterodimerization, it sequesters the full-length receptor inside the cell. The loss of the truncated receptor isoform caused by oligo#5 releases this sequestration and causes an accumulation of the receptor on the cell surface, which results in an anorexic response.

Data information: (B–D) Error bars indicate standard deviation. Student's *t*-test.

cDNA analysis. A potential physiological role for the truncated receptor isoform emerged only recently. The truncated isoform is localized to internal membranes, and through heterodimerization, it sequesters the full-length receptor in intracellular sites, reducing serotonin receptor signaling (Martin *et al*, 2013). Using cell fractionation, we provide additional evidence for this model, as we found that the full-length receptor is only detectable in membrane fractions, whereas the truncated receptor also is present in the cytosol, indicating a weaker association with membranes. Through heterodimerization, the short splicing variant acts like a dominant negative modulator of signaling through the 5HT2C receptor. Regulating the balance between truncated and full-length isoforms therefore changes the physiological response of the receptor. Oligo#5 reduces the amount of the truncated isoform and increases the amount of the full-length receptor protein. The oligo thus changes the relative ratio of the two proteins, without completely abolishing the production of the truncated receptor isoform. Using an EGFP-tagged splicing reporter resembling the physiological system, we observed accumulation of full-length receptor on the cell surface in cultured cells after oligo#5 treatment, which likely increase serotonin signaling.

We further tested this model in mice by injecting oligo#5 into the third ventricle. The arcuate nucleus that harbors POMC-positive neurons expressing the 5HT2C receptor is naturally open toward the cerebrospinal fluid and allows, for example, entry of peptide hormones, such as insulin and ghrelin (Rodriguez *et al*, 2010). As we observe accumulation of oligo#5 in the arcuate nucleus, it is likely that oligo#5 follows the same uptake route as peptide hormones. Oligo#5 injection increased RNA2, as predicted from the transfection experiments and increased POMC expression. This could explain the observed reduction in food intake, as POMC is processed to α-melanocyte-stimulating hormone (α-MSH), resulting in an anorexic response by mainly activating melanocortin 4 receptor-containing neurons (MCR4) in the paraventricular nucleus. In agreement with this model, oligo#5 had no statistically significant effect in MCR4 knockout mice (Srisai *et al*, 2011), (Fig EV5). However, since the full-length 5HT2C receptor heterodimerizes with other seven-transmembrane receptors (Schellekens *et al*, 2013, 2015), it is possible that oligo#5 affects the surface localization of unknown receptor systems, which could contribute to the anorexic response. Oligo#5 had no effect on water uptake or general activity of the animals (Fig EV6), suggesting that the oligo modifies a specific pathway, rather than causing a general sickness that stops food intake.

Finally, we tested oligo#5 by systemic injection, using mice with a catheter in the common carotid artery. Low dosages of oligo#5 showed no statistically significant effect on food uptake (data not shown), but injection of 50 or 100 µg reduced food intake. As the effect on food uptake was unexpected, we repeated the experiments using Cy3-labeled oligonucleotide and observed rapid accumulation of the oligonucleotide in the brain. The strongest accumulation was in the choroid plexus, a structure that generates cerebrospinal fluid from blood (Mortazavi *et al*, 2013). Since the choroid plexus expresses high levels of RNA1, it is possible that oligo#5 binds to 5HT2C pre-mRNA expressed in ependymal cells and follows the secreted CSF. This transport would follow the concentration gradient between blood and CSF and is facilitated by the ability of oligo#5 to enter cells without adjuvants. In this model, the 5HT2C pre-mRNA acts like a receptor for oligo#5. It is possible that the known *in vivo* secondary structure of the receptor helps in this interaction, as it is likely more stable than unstructured pre-mRNAs.

Similar to oligo#5, SNORD115 promotes inclusion of exon Vb into the pre-mRNA, reducing the amount of the truncated receptor isoform. SNORD115 is absent in subjects with PWS, suggesting that a deregulation of the 5HT2C pre-mRNA splicing, in particular, an increase of the truncated receptor isoform, contributes to PWS. As the full-length 5HT2C heterodimerizes with other receptors, such as the ghrelin receptor (Schellekens *et al*, 2013), it could indirectly regulate other receptors involved in food uptake, which would amplify the effect of the truncated isoform. Thus, a deregulation of 5HT2C splicing caused by the loss of SNORD115 could contribute to a hormonal syndrome seen in PWS. The basis of this regulation is a stable double-stranded RNA structure in the 5HT2C pre-mRNA. This structure can be targeted by small-molecular-weight substances like pyrvinium pamoate (Shen *et al*, 2013) or as we showed here, an oligonucleotide. The 5HT2C pre-mRNA thus forms a promising drug target to treat hyperphagia seen in PWS and possibly to treat non-genetic forms of obesity.

## Materials and Methods

### Cell culture and transfection

HEK293T cells (ATCC) were cultured in DMEM containing 10% (v/v) fetal bovine serum (Invitrogen). RNA oligonucleotides and plasmid DNA were transfected into cells with calcium phosphate as

described previously (Kishore *et al*, 2010). About 50 ng of RNA nucleotide and 100 ng of DNA plasmid were transfected into $1 \times 10^6$ cells. Serotonin 2C receptor pre-mRNA splicing analysis was performed as described (Kishore & Stamm, 2006).

## Generation of stable cell lines

An established Flp-in HeLa cell line was a gift from Dr. Jørgen Kjems (Aarhus University, Denmark). The generation of a Flp-in HeLa cell line stably expresses GFP-5HT2C as described in Damgaard *et al* (2012). The stable cells were maintained in medium as described previously with 600 μg/ml hygromycin.

## Construction of plasmids

The serotonin 2C receptor (5HT2C) splicing reporter construct used in transfecting cultured cells was described before (Kishore & Stamm, 2006). To generate a reporter construct for *in vitro* splicing, a DNA fragment including exon V (201nt) and 5′ end of intron V (177 nt) from the above construct was PCR amplified with primers 5′-Exon5(kpnI) and 3′-Intron5(BamH1). The fragment was then used to replace the first exon in MINX splicing reporter gene.

To generate 5HT2C fused with a C-terminal GFP construct, the first DNA fragment containing exon 3, exon 4, and partial exon 5 was PCR amplified from human 5HT2C cDNA construct with primers 5′-Exon3(NheI) and 3′-Exon5(BglII). The fragment was then subcloned into pEGFPN3. The second DNA fragment includes the rest of exon 5 and partial intron 5 (1.5 kb from 5′ end). It was PCR amplified from HTR2C splicing reporter construct (Kishore & Stamm, 2006) with primers 5′-Exon5(BamH1) and 3′-Exon6. The PCR product was then subcloned with BamH1 and BglII following the first fragment into pEGFPN3. The third fragment includes 3′ end of intron V (1.6 kb) and the encoding sequence in exon VI. It was PCR amplified with primers 5′-Intron5(Bglll) and 3′Exon6(BamH1). The third fragment was inserted into pEGFP3 after the second fragment.

For stable cell line generation with Flp-in system, the 5HT2C–GFP fusion fragment described above was subcloned between NheI and NotI into the pcDNA5_FRT construct (Invitrogen Flp-in system. DNA constructs are available from Addgene, Cambridge, MA, Deposit number 73069.

## *In vitro* splicing assay

[32]P-labeled pre-mRNAs were *in vitro* transcribed using an SP6 polymerase after linearizing the HTR2C-MINX construct. The substrates were incubated at 30°C with 40% HeLa nuclear extracts under standard splicing conditions as described in (Mayeda & Krainer, 2012). The RNA splicing products and intermediates were separated on 5% denatured polyacrylamide gels and visualized with a phosphorImager (GE Typhoon). The sequence of the *in vitro* splicing construct was as follows: Exons are in capital letters, the SP6 promoter is in italic, the transcriptional start site is in capital and bold, and the intronic and exonic part of the adenovirus derived MINX construct is underlined. The oligo#5 binding site is double underlined.

*gtaatacgactcactata***G***gg*cgaattgggtaccATTATGTCTGGCCACTACCT AGATATTTGTGCCCCGTCTGGATTTCTTTAGATGTTTTATTTTCAA CAGCGTCCATCATGCACCTCTGCGCTATATCGCTGGATCGGTATGT AGCAATACGTAATCCTATTGAGCATAGCCGTTTCAATTCGCGGAC

TAAGGCCATCATGAAGATTGCTATTGTTTGGGCAATTTCTACAGgt aagtaaaacttttttggccataagaattgcagcggctatgctcaatactttcggattatgtactgtgaac aacgtacagacgtcgactggtaacatttgcgtttgggatctgctgcacgtctagggcgcagtagtcc aggggtttccttgatgatgtcatacttatcctgtccctttttttttccacagCTCGCGGTTGAGGA CAAACTCTTCGCGGTCTTTCCAGTGGGGATCC

## Development of an antiserum against RNA1

Rabbits were immunized using Ac-SSYPCDWTEGRRKGV-Ahx-[K-HZ]-amide; Ac-SSYPCDWTEGRKQSV-Ahx-[K-HZ]-amide; HZ-Ahx-SSYPCDWTEGRRKGV-amide and HZ-Ahx-SSYPCDWTEGRKQSV-amide. The underlined sequence SSYPCDWTEGR is common to both mouse and human, and the remaining RKGV are specific for human and KQSV are specific for mouse. After six immunizations, the last three boosts containing complete Freund's adjuvants, an antiserum specific for RNA1 was obtained, which as affinity purified using the immunization peptide.

## Antibodies used

Anti-GFP was from Santa Cruz, sc-8334 (rabbit polyclonal), NeuN from Millipore, MAB377, clone A60. All antibodies were used at a 1:1,000 dilution in PBS/0.1% Tween-20.

## Detection of serotonin 2C receptor isoforms using Western Blot

About $5 \times 10^5$ HeLa cells that stably express GFP-5HT2C were seeded into 6-well plates. The next day, control oligo#3 and oligo#5 were added to the cells at a 5 nM final concentration. The cell fractionation into membrane and cytosolic fractions was according to Mancia *et al* (2008). Forty-eight hours after transfection, the cells were washed with cold PBS and were lysed in CHS buffer (20 mM Hepes pH 7.5, 150 mM NaCl, and 0.1% w/v cholesteryl hemisuccinate in 1% w/v n-dodecyl-β-d-maltopyranoside, Sigma) for 30 min on ice. The soluble fraction, enriched in cytosol, was transferred to a fresh tube after centrifugation of 10 min at 13,500 *g*, 4°C. The insoluble fraction enriched in membranes was dissolved in 1× SDS loading buffer. About 20 μg of protein was loaded on a 12% SDS–PAGE gel.

## Cell imaging

Stable GFP-HTR2c cells were seeded onto sterilized cover slips placed to the bottom of tissue culture plates. Oligo#5 was then added to final concentration of 5 nM. Oligo#3 was used as control. After 48 h, the cells were washed with PBS and fixed with 4% paraformaldehyde for 5 min. The cells were then incubated for 10 min with PBS containing 0.25% Triton X-100 and stained with DAPI. The cells on cover slips were analyzed by confocal microscopy (Nikon A1R-A1 Confocal AQ5 Microscope System).

## Mouse injection and food uptake measurement

Adult male mice (9–12 weeks old) were acquired from Jackson Laboratory. All mice were maintained on a 14/10-h light/dark cycle with food and water provided *ad libitum*. Animal work was approved by the IACUC from the University of Kentucky Protocol Number: 2011-0841.

The mice were stereotaxically implanted with a guide cannula (Plastics One) to the coordinates of 1.82 mm posterior to the

Bregma, 4.5 mm below the surface of the skull, and 0 mm lateral to midline. The cannula tip is located above the 3rd ventricle. After the surgery, the mice were individually housed and deprived of food with continuous water supply. The injection was performed 24 h after cannula implant. An infusion cannula with 1.0 mm projection (Plastics One) was inserted through the guide cannula and used to infuse 2 μl (1 μg/μl) of RNA nucleotides over 2 min. Feeding was resumed after the injection. The body mass and food intake were recorded at the designated time points.

Nine-week-old C57BL/6 mice with a polyurethane/polyethylene carotid artery catheter were purchased from Charles River, (Charles River, INC). The catheter was inserted into the common carotid artery with the catheter tip directed toward the brain. About 50–100 μg oligonucleotide (10 μg/μl) was infused over 3–5 min. Dosages lower than 31 μg showed no significant effect (1, 2, 10, 20, and 30 μg were tested).

Ob/ob mice B6.Cg-*Lep^{ob}*/J were acquired from the Jackson Laboratory, Bar Harbor, Maine.

### Statistical analysis

All the data obtained were from at least three independent experiments. Data were analyzed with Microsoft Excel version 14.0.7166.5000 (32-bit) and expressed as the mean ± SD (standard deviation based on the entire population given as arguments, STDEVP function in Excel). The statistical differences were calculated using Student's *t*-test, and the difference was considered to be statistically significant at values of $P \leq 0.05$. The number of samples analyzed and the *P*-value are indicated in the legends of each figure.

### qPCR

Three micrograms of total RNA, 2 μM oligo-dT, 1 mM dNTPs, and $H_2O$ were mixed and heated to 65°C for 5 min and quickly chilled on ice. 1× Reverse transcription buffer and 20 mM of DTT were added to the reaction. The reaction was incubated 42°C for 2 min and 200 U of SuperScript II reverse transcriptase was subsequently added to the reaction, in a total volume of 20 μl. The reverse transcription was performed under the following conditions: 50 min at 42°C and 15 min at 70°C. One-tenth of the RT reaction was used for the qPCR reaction. The reaction was performed in 20 μl that contained 1× SYBR Green SuperMix (Quanta Bioscience) and 0.5 mM of gene-specific primers. The amplification was carried out in a Stratagene Mx3005P Thermocycler (Agilent Technologies) using the following conditions: initial denaturation for 10 min at 95°C, 40 cycles for 15 s at 95°C, and an extension of 1 min at 60°C. The relative expression was estimated as follows: $2^{Ct(\text{reference}) - Ct(\text{sample})}$, where $C_t$ (reference) and $C_t$ (sample) were GAPDH and target genes from same RT reaction, respectively. For each experiment, at least three animals were analyzed.

qPCR primers
gapdh
mGAPDHFor1: TCGTCCCGTAGACAAAATGG
mGAPDHRev1: TTGAGGTCAATGAAGGGGTC

RNA2
Htr2cEx4 For1: ACTTGTCATGCCCCTGTCTC
mHtr2cExVb Rev1: CCTTAGTCCGCGAATTGAAC

RNA1
mHtr2cEx4 F: CCATTGCTGATATGCTGGTG
mRNA1junc R: ACTGAAACTCCCGGTCCAG

POMC primers:
mPOMC-F1: GAGTTCAAGAGGGGAGCTGGA
mPOMC-R1: CTTGATGATGGCGTTCTTGA

Serotonin receptor Htr4
mHtr4-F1: TGCTATCACCTGCTCTGTGG
mHtr4-R1: CTGCCTTGGTCTCTGTCCTC

### RNA antisense oligonucleotide synthesis

Complementary 18-base RNA antisense oligonucleotides targeting 5HT2C gene exon 5 and intron 5 junction region for the RNA oligo walk were synthesized and HPLC-purified from Trilink BioTechnologies (San Diego, CA). All the nucleotides were 2′-O-methyl-modified bases with a monophosphothioate backbone, containing one sulfur atom per phosphate group. The oligonucleotide sequences were as follows:

OLIGO #1: 5′ UAAUCCGAAAGUAUUGAG 3′
OLIGO #2: 5′ CAUAGCCGCUGCAAUUCU 3′
OLIGO #3: 5′ UAGCAAUCUUCAUGAUGG 3′
OLIGO #4: 5′ UUAGUCCGCGAAUUGAAA 3′
OLIGO #5: 5′ AGUAUUGAGCAUAGCCGC 3′
OLIGO #5-5: 5′ AGUAUUGAGCAU 3′
OLIOG #5-3: 5′ GAGCAUAGCCGC 3′
OLIGO #5-10: 5′ GCAUAGCCGC 3′
OLIGO #6: 5′ UGCAAUUCUUAUGGCCAA 3′
OLIGO #7: 5′ AUGCUCAAUAGGAUUACG 3′
OLIGO #8: 5′ AGAAAUUGCCCAAACAAU 3′
OLIGO #9: 5′ AAAGUUUUAUUUACCUAU 3′
SMN2: 5′-AUUCACUUUCAUAAUGCUGG-3

### List of primers for PCR

To clone of HTR2C-MINX construct

5′-Exon5(kpnI): CCTCGGTACCATTATGTCTGGCCACTACC
3′-Intron5(BamH1):CCATTAGGATCCTTTCATTATTGTTAACC

To clone the pEGFPN3-HTR2C construct

5′-Exon3(NheI): CATACGGCTAGCCACCATGGTGAACCTGAGG
3′-Exon5(BglII): CTTGGCAGATCTAGGTAGTGGCCAGAC
5′-Exon5(BamH1): CATACGGGATCCTTGTGCCCCGTCTGGATTTC
3′-Exon6: CGTCCCTCAGTCCAATCACAG
5′-Intron5(BglII): CTAGAGCGGCCGCATCAAGATCCC
3′-Exon6(BamH1): CTTAGCGGATCCCACACTGCTAATCCTTTCGC

**Expanded View** for this article is available online.

### Acknowledgements

We are very grateful to the NIMH/Vanderbilt Silvio O. Conte Center for support of these studies (P50 MH096972). This work also was supported by National Institutes of Health RO1 GM083187, R21HD080035, and P20RR020171 to S. S and the Foundation for Prader–Willi Research, and by an Institutional Development Award (IDeA) from the National Institute of

## The paper explained

### Problem

Obesity is a growing health problem, caused by a behavior resulting in caloric intake that exceeds caloric expenditure over longer time periods. Prader–Willi syndrome is a genetic form of obesity, caused by hyperphagia and the inability to reach satiety. The serotonin 2C receptor in proopiomelanocortin (POMC) neurons of the hypothalamic arcuate nucleus plays an important role in food uptake behavior. Serotonin 2C receptor activation results in an induction of POMC, which is processed into α-melanocyte-stimulating hormones (α-MSH, member of the melanocortin peptide family). α-MSH activates the melanocortin 4 receptor (MC4R) in the paraventricular nucleus, leading to an inhibition of food uptake. The serotonin 2C receptor undergoes pre-mRNA splicing resulting in a full-length receptor and a truncated receptor, depending on the usage of the alternative exon Vb. In contrast to the full-length receptor that is present on the cell surface, the truncated receptor is found only intracellularly and can sequester the full-length serotonin 2C receptor. In addition to alternative splicing, the serotonin 2C receptor pre-mRNA undergoes RNA editing, which changes the protein sequence of the full-length receptors. The non-edited receptor shows constitutive activity, which is strongly reduced by RNA editing. The inclusion of exon Vb resulting in a non-edited full-length receptor is promoted by a neuron-specific trans-acting RNA, SNORD115. SNORD115 is not expressed in subjects with Prader–Willi syndrome. As a possible therapeutic approach, we sought to find a substitution strategy for SNORD115 and tested its effect on food uptake in mice.

### Results

Using oligonucleotides complementary to the regulated part of the serotonin 2C receptor pre-mRNA, we identified an oligonucleotide (oligo#5) that promoted inclusion of exon Vb into the pre-mRNA, thereby increasing the full-length, non-edited receptor isoform. Oligo#5 binds to an intronic region and will thus not interfere with translation. Injection of oligo#5 into mouse brain increased exon Vb usage and POMC mRNA concentration in the arcuate nucleus, which likely caused the subsequent reduction in food uptake. Using a new antiserum, we showed that the truncated serotonin 2C receptor is expressed as a protein and that oligo#5 increases the ratio of full-length to truncated receptor proteins, leading to an accumulation of the constitutively active full-length receptor at the cell surface. Unexpectedly, delivery of oligo#5 through the carotid artery resulted in its accumulation in the brain, an increase of the full-length receptor isoform and a reduction in food uptake that persisted for 3 days after injection.

### Impact

The serotonin 2C receptor is a validated anti-obesity drug target, but similarities with other serotonin receptors on the protein level prevented widespread use of its protein agonists as appetite suppressors. Targeting its RNA through oligonucleotides could be a viable alternative, as the oligo#5 sequence is unique in the genome and thus will be more selective. The oligo#5-induced changes in serotonin 2C receptor splicing isoforms had a profound effect on food intake, which underscores the role of alternative splicing variants in food regulation. Since the serotonin 2C receptor heterodimerizes with other G-coupled protein receptors, the regulation of its surface concentration through alternative splicing could impact on other receptors, which could help in integrating food uptake with other behavioral programs. Finally, oligo#5 mimics the action of the natural exon Vb activator, SNORD115, which is not expressed in Prader–Willi syndrome. This points to a deregulation of the serotonin receptor isoforms in Prader–Willi syndrome caused by SNORD115 loss as an underlying cause of the characteristic hyperphagia.

General Medical Sciences of the National Institutes of Health (8 P20 GM103527-06).

## Author contributions

ZZ and MS performed the cloning experiments and injections; PJG, AGR, and MGL assisted in the mouse work; RBE and SS planned the experiments.

## Conflict of Interest

The authors declared that they have no conflict of interest.

## For more information

https://www.fpwr.org/

http://www.pwsausa.org/

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
