## [Review Process File · EMBO Molecular Medicine]

Oligonucleotide-induced alternative splicing of serotonin 2C receptor reduces food intake

Zhaiyi Zhang, Manli Shen, Paul Gresch, Masoud Ghamari-Langroudi, Alexander G. Rabchevsky, Ronald Emeson and Stefan Stamm

Corresponding author: Stefan Stamm, University of Kentucky

Review timeline:	Transfer date from The EMBO Journal:	03 November 2015
	Editorial Decision:	02 December 2015
	Revision received:	04 May 2016
	Editorial Decision:	23 May 2016
	Revision received:	27 May 2016
	Accepted:	09 June 2016

Transaction Report:

Editor: Céline Carret

1st Editorial Decision

02 December 2015

Thank you for the submission of your manuscript to EMBO Molecular Medicine. We have now heard back from the two referees whom we asked to evaluate your manuscript. Although the referees find the study to be of potential interest, they also raise a number of concerns that need to be addressed in the next version of your article.

You will see that while both referees find the study of significant interest they do have several suggestions that if followed, would strongly strengthen the study. Of particular importance, controls and additional experiments should be added to increase the conclusiveness as well as clarifications and better description of the in vivo data should be provided. I would like take the opportunity to say that statistics and animal data are subjected to strict guidelines in all EMBO Press publications and that you must comply and provide a checklist detailing statistical analyses, number of replicates etc, as well as detailed experimentation on animals, and appropriate licensing to do so (please see point 8 below).

It is our opinion that all suggested experiments and text modifications are reasonable and would improve the impact of the paper and I would therefore encourage you to address these in a major revision of your work. Please note that it is EMBO Molecular Medicine policy to allow only a single round of revision and that, as acceptance or rejection of the manuscript will depend on another round of review, your responses should be as complete as possible.

EMBO Molecular Medicine has a "scooping protection" policy, whereby similar findings that are published by others during review or revision are not a criterion for rejection. Should you decide to submit a revised version, I do ask that you get in touch after three months if you have not completed

it, to update us on the status.

I look forward to seeing a revised form of your manuscript as soon as possible.

***** Reviewer's comments *****

Referee #1 (Remarks):

In the manuscript, Zhang et al., developed and characterized a series of RNA oligonucleotides that target the splicing of the serotonin 2C receptor pre-mRNA resulting in the selective production of the full-length transcripts against the truncated isoforms. Using in vitro cell culture systems, the authors demonstrated that these oligos were able to modify the alternative splicing of reporter constructs for the serotonin 2C receptor exons and may subsequently alter its protein synthesis in transfected cells. The authors further showed that these oligos can be targeted to the mouse brain by intracerebroventricular and systematic deliveries. In both cases, oligo treatments can acutely suppress food intake in wildtype and in obese ob/ob mice.

The data presented are quite novel and important. The serotonin 2C receptor is the target of an FDA-approved weight loss agent, whose malfunction may contribute to the pathologies of the Prader-Willi syndrome. As a result, findings in the paper is clinical relevant. Moreover, the authors presented a large amount of data in the manuscript, and the use of a combination of genetic, histological, and behavioral analyses is a major strength of this work. Nevertheless, the current manuscript still needs improvement. The biggest criticism of the manuscript is that this work though quite interesting relies mainly on experiments performed using artificial mini genes/constructs in culture systems that do not express the serotonin 2C receptors. The effect on splicing of the endogenous gene is less clear. The lack of supporting evidence at the protein level, especially in mice, is another pitfall. Similarly, the in vivo feeding studies appear to be incomplete.

Figure 3. The authors' claim "oligo#5 promotes exon Vb inclusion without RNA editing" was solely based on the results by Sanger sequencing. There was no positive control for RNA-edited transcript in this experiment. RNA editing could still occur in some cells; however, reverse transcription could disproportionally amplify the predominant unedited transcripts leading to the negative findings. The authors need collaborate their findings with deep-sequencing or other suitable approaches for detecting RNA editing.

Figure 4. It appears that the cells in the control group were still able to produce some gfp-tagged full-length serotonin 2C receptors. However, the gfp signals were mostly found in the cytoplasm. Should they be targeted to the cell surface as well? Is the change in surface distribution merely the result of alterations in protein levels? Should this be the case, the authors could simply use western blot to quantify the protein levels of gfp (a surrogate marker for the full length protein), which would provide stronger support for their conclusion.

Figure 5. It will be better to use antibodies against POMC (-MSH) to show that the cy3-tagged oligos can directly target POMC neurons. The reduction in food intake is rather striking in wildtype and in ob/ob mice. Changes in POMC alone probably won't produce such an effect. On the other hand, there was no description of any longer-term effect on feeding beyond 12 hours. It is therefore unknown whether the reduction in food intake in these animals was caused by a suppression of appetite or due to other sickness behaviors.

Figure 6. The wide distribution of oligos following systematic injection is not consistent with the findings in the icv experiments which showed limited penetration of the oligos. Again, there is no longer term data beyond 12 hours.

Minor:

The manuscript contains several typos and grammatical errors and could use better proofreading.

Referee #2 (Remarks):

The manuscript entitled "Changing serotonin receptor 2C alternative splicing with an oligonucleotide reduces food intake" by Zhang et al nicely builds on previous work from the Stamm lab showing that SNORD115 regulates splicing of the 5HT2C pre-mRNA. Here an oligonucleotide is elegantly utilized to modify splicing of the 5HT2C pre-mRNA to support the full-length functional transcript to reinstate healthy food intake. The in vitro and in cellulo findings are supported by strong animal data. The work has important implications for therapy of obesity due to overeating in genetic diseases such as Prader-Willi syndrome in which SNOR115 is deregulated as well as in non-genetic forms of obesity.

Scientific and/or conceptual concerns:

Though most of the RT-PCR assays to detect splicing changes include quantitation and statistical analysis, Figures 2C and 4B do not. Have these experiments been repeated? Are they reproducible? Are the changes seen statistically significant?

Has the in vitro splicing assay depicted in figure 3D been repeated? Is it reproducible? Are the changes seen statistically significant?

The authors state "Oligo#5 acts directly on the pre-mRNA, likely without involvement of chromatin structures or other cellular proteins." Though the in vitro splicing assays suggest that chromatin is not involved, there is no experimental evidence to support the claim that other cellular proteins are not involved and the statement should be modified accordingly.

The authors show nicely that oligo#5 was taken up by cells without adjuvants. The authors further show that oligo#5 is detected in neurons by co-staining with a neuronal marker and state that the oligo is also in a cell type thought to be glial cells. Co-staining with a glial marker should be used to make this claim.

It is not clear why the control oligo utilized (SMN2) was chosen. The reason for using the SMN2 oligo should be discussed. Does the SMN2 oligo used confer splicing changes in SMN? Is it known to also enter neuronal cells? Is it a similar length and nucleotide composition as Oligo#5?

In reference to the data shown in Figure 5, why are the effects on feeding behavior longer lived than the splicing changes? This should be discussed.

For the experiments using oligo delivery via the blood, were the splicing changes assessed? This data seems integral to the paper and should be included.

The authors state that the restoration of splicing and full-length receptor likely increases serotonin signaling. Testing serotonin signaling should be straightforward would strengthen the claims and impact of this manuscript.

Minor comments:

In the abstract, it should be made clear that the injection of oligonucleotide into the third ventricle was made in mice.

The chemistry of the oligo's used in the Passini 2011 paper differed from the chemistry of the oligo's used here. It is cited that the oligo's had similar chemistry, but the differences should be directly addressed.

In figures 3A and 3B, it is not clear what the boxes below the sequencing trace data signify. Presumably the 5 edited sites? This information should be included in the figure legend.

RNA1 and RNA2 are discussed in reference to Figure 5, yet the identity of these RNAs is not defined until the model in Figure 6D. The nomenclature should be modified or RNA1 and 2 defined earlier in the paper.

In the discussion, the authors state that they sought "substitution" therapies. This has an unclear meaning. Do they perhaps mean "alternative" therapies?

Also in the discussion section, it is stated that the oligos show no RNaseH activity. Since RNA oligos don't have RNaseH activity, but can be degraded by RNAase H, the wording should be modified for clarity.

The Materials and Methods section (incorrectly labeled Material and Methods) is lacking details about dosing for the stereotaxically treated animals. This section is likewise lacking any detailed description of the systemic injection of oligo via the catheterized carotid artery. Additionally, low doses that had no statistically significant effect (data not shown) were not disclosed at all.

In the figure 5B legend, there is a typographical error: acruate should be arcuate.

1st Revision - authors' response

04 May 2016

Referee #1 (Remarks):

In the manuscript, Zhang et al., developed and characterized a series of RNA oligonucleotides that target the splicing of the serotonin 2C receptor pre-mRNA resulting in the selective production of the full-length transcripts against the truncated isoforms. Using in vitro cell culture systems, the authors demonstrated that these oligos were able to modify the alternative splicing of reporter constructs for the serotonin 2C receptor exons and may subsequently alter its protein synthesis in transfected cells. The authors further showed that these oligos can be targeted to the mouse brain by intracerebroventricular and systematic deliveries. In both cases, oligo treatments can acutely suppress food intake in wild type and in obese ob/ob mice.

The data presented are quite novel and important. The serotonin 2C receptor is the target of an FDA-approved weight loss agent, whose malfunction may contribute to the pathologies of the Prader-Willi syndrome. As a result, findings in the paper is clinical relevant. Moreover, the authors presented a large amount of data in the manuscript, and the use of a combination of genetic, histological, and behavioural analyses is a major strength of this work. Nevertheless, the current manuscript still needs improvement.

The biggest criticism of the manuscript is that this work though quite interesting relies mainly on experiments performed using artificial mini genes/constructs in culture systems that do not express the serotonin 2C receptors. The lack of supporting evidence at the protein level, especially in mice, is another pitfall.

We raised a peptide antibody specific for the truncated receptor isoform (Figure 1A) and show that this antibody recognizes the predicted protein in brain (Figure 1B). The changes predicted by the mRNA changes are reflected on the protein level in cell culture (Figure 4C-E).

We add to the text: Whereas the full-length 5HT2C is 90% identical between mouse and human and differs only by one amino acid in length, the C-terminus of the truncated receptor contains 96 amino acids in humans, but only 19 amino acids in mouse (Figure EV1A). Since the stop codon generating the truncated receptor is located in the last exon, its mRNA is likely not subject to nonsense-mediated decay (Fatscher et al, 2015). To test whether RNA2 is expressed as an endogenous protein, we generated an antiserum against a peptide common to mouse and human RNA2 (Figure EV1B). After affinity purification, this antiserum detects truncated 5HT2C protein from HEK293 cells transfected with a cDNA encoding the truncated receptor, as well as the endogenous protein from mouse brain (Figure 1B), which shows the existence of a truncated 5HT2C in vivo.

The effect on splicing of the endogenous gene is less clear.

The changes in splicing were measured using real-time PCR, both after ICV injection and after carotid injection (Figure 5E, Figure 6B), which shows the changes in isoform ratios.

Similarly, the in vivo feeding studies appear to be incomplete.

We extended the feeding studies for up to a week. Most mice start to pull out the catheters after

three days, leading to complications. We thus show the data for three days (Figure 6D), that support the efficacy of oligo#5 and add to the text: The effect of oligo#5 injection persisted over three days (Figure 6D).

Figure 3. The authors' claim "oligo#5 promotes exon Vb inclusion without RNA editing" was solely based on the results by Sanger sequencing. There was no positive control for RNA-edited transcript in this experiment. RNA editing could still occur in some cells; however, reverse transcription could disproportionately amplify the predominant unedited transcripts leading to the negative findings. The authors need collaborate their findings with deep-sequencing or other suitable approaches for detecting RNA editing.

We repeated the experiment in brain using NGS sequencing approach. Again, there was no change in editing. We add these data as a Figure 5F.

Figure 4. It appears that the cells in the control group were still able to produce some gfp-tagged full-length serotonin 2C receptors. However, the gfp signals were mostly found in the cytoplasm. Should they be targeted to the cell surface as well? Is the change in surface distribution merely the result of alterations in protein levels? Should this be the case, the authors could simply use western blot to quantify the protein levels of gfp (a surrogate marker for the full length protein), which would provide stronger support for their conclusion.

Using our new antiserum, the experiments suggested by the reviewer were performed and added to the manuscript (Figure 4C-E). We also clarify in the discussion that the oligo changes the ratio of the protein isoforms and add to the text:

The oligo thus changes the relative ratio of the two proteins, without completely abolishing the production of the truncated receptor isoform.

Figure 5. It will be better to use antibodies against POMC (α-MSH) to show that the cy3-tagged oligos can directly target POMC neurons. The reduction in food intake is rather striking in wild type and in ob/ob mice. Changes in POMC alone probably won't produce such an effect. On the other hand, there was no description of any longer-term effect on feeding beyond 12 hours. It is therefore unknown whether the reduction in food intake in these animals was caused by suppression of appetite or due to other sickness behaviours.

The long-term effect on feeding was added as Figure 5H. Staining of alpha MSH in POMC neurons was technically not feasible due to the low number of 1000 POMC neurons and the required double/triple staining. We therefore tested the effect of oligo#5 in melanocortin 4 receptor knock out mice. The MCR4 senses alpha MSH and its activation triggers the anorexic response. We found no statistical significant effect of oligo#5 on these mice.

The suppression of appetite/sickness, was determined by monitoring the movement of the animals, measured by accumulated beam activity, which is added as Figure EV5.

We add to the text:

In agreement with this model, oligo#5 had no statistically significant effect in MCR4 knock out mice (Srisai et al, 2011), (Figure EV5). However, since the full-length 5HT2C heterodimerizes with other seven-transmembrane receptors (Schellekens et al, 2015; Schellekens et al, 2013), it is possible that oligo#5 affects the surface localization of unknown receptor systems, which could contribute to the anorexic response. Oligo#5 had no effect on water uptake or general activity of the animals (Figure EV6), suggesting that the oligo modifies a specific pathway, rather than causing a general sickness that stops intake.

Figure 6. The wide distribution of oligos following systematic injection is not consistent with the findings in the icv experiments, which showed limited penetration of the oligos.

The difference in oligo#5 distribution is likely due to the amount used (1-2 µg/vs 50-100µg) as well as the delivery through choroid plexus.

We add to the text:

Delivery through carotid injection results in a more widespread distribution of oligo#5 in the brain than ICV injection into the third ventricle. This is likely due to the 50-100 fold higher amount of oligo#5 injected and could reflect oligo entry through each of the choroid plexuses.

Referee #2 (Remarks):

The manuscript entitled "Changing serotonin receptor 2C alternative splicing with an oligonucleotide reduces food intake" by Zhang et al nicely builds on previous work from the Stamm lab showing that SNORD115 regulates splicing of the 5HT2C pre-mRNA. Here an oligonucleotide is elegantly utilized to modify splicing of the 5HT2C pre-mRNA to support the full-length functional transcript to reinstate healthy food intake. The in vitro and in cellulo findings are supported by strong animal data. The work has important implications for therapy of obesity due to overeating in genetic diseases such as Prader-Willi syndrome in which SNOR115 is deregulated as well as in non-genetic forms of obesity.

Scientific and/or conceptual concerns:

*Though most of the RT-PCR assays to detect splicing changes include quantitation and statistical analysis, **Figures 2C and 4B do not.** Have these experiments been repeated? Are they reproducible? Are the changes seen statistically significant?*

We added the statistical analysis as Figures 2D and 2F, all effects were significant (p<0.001-0.1).

*Has the in vitro splicing assay depicted in figure 3D been repeated? **Is it reproducible?** Are the changes seen statistically significant?*

The in vitro experiments have been performed numerous times, as three exon-substrates are difficult to splice. We quantified the result for exonVb inclusion, p<0.01, shown as Figure 3E.

*The authors state "Oligo#5 acts directly on the pre-mRNA, likely without involvement of chromatin structures or other cellular proteins." Though the in vitro splicing assays suggest that chromatin is not involved, there is no experimental evidence to support the claim that other cellular proteins are not involved and the statement **should be modified accordingly.***

Thanks for pointing this out. Since this is a minor point, we omit this sentence.

The authors show nicely that oligo#5 was taken up by cells without adjuvants. The authors further show that oligo#5 is detected in neurons by co-staining with a neuronal marker and state that the oligo is also in a cell type thought to be glial cells. Co-staining with a glial marker should be used to make this claim.

We added GFAP staining as Figure 5D and observe staining of some, but not all glial cells.

We add to the Figure:

D. Oligo#5 enters some glia cells. Enlargement of a hypothalamic region adjacent to the 3rd ventricle one hour after oligo#5 injection. The oligo (red) can be found in some GFAP-positive glia cells (arrow), but numerous GFAP-positive cells show no oligo#5 uptake.

It is not clear why the control oligo utilized (SMN2) was chosen. The reason for using the SMN2 oligo should be discussed. Does the SMN2 oligo used confer splicing changes in SMN? Is it known to also enter neuronal cells? Is it a similar length and nucleotide composition as Oligo#5?

In our injection strategy, we followed the approach of the SMA and Duchenne muscular dystrophy

field, initially using a Cy3 SMN2 oligo (2'-O-methyl-phosphothioate) to monitor uptake and developed oligo#5 using this chemistry.

We clarified this point in the text:

In all injection experiments, we used an oligonucleotide against human SMN2 as a control (Seo et al, 2014). The SMN2-20mer oligonucleotide had the same 2'-O-methyl mono-phosphothioate chemistry as oligo#5. Oligonucleotides with this chemistry are taken up by cell without adjuvans (Heemskerk et al, 2009), and SMN2 oligonucleotides with the related 2'-O-(2-methoxyethyl) phosphorothioate chemistry were previously shown to enter spinal cord neurons after epidural injection in mice (Hua et al, 2010). SMN2 is not expressed in mice and thus the oligonucleotide has no specific target and controls for unspecific effects of injection.

In reference to the data shown in Figure 5, why are the effects on feeding behaviour longer lived than the splicing changes? This should be discussed.

We extended the feeding studies for up to a week. Most mice start to pull out the catheters after three days, leading to complications. We thus show the data for three days (Figure 6D), that support the efficacy of the oligo over a longer time.

For the experiments using oligo delivery via the blood, were the splicing changes assessed? This data seems integral to the paper and should be included.

We performed the suggested experiments after injection, which is shown as Figure 6B.

To determine a possible effect on 5HT2C processing, we micro dissected the region around the 3rd ventricle and analysed 5HT2C splicing using real-time PCR (Figure 6B). Again, we found an increase of the RNA2/RNA1 ratio, suggesting that oligo#5 influences 5HT2C splicing after crossing the blood-brain barrier.

The authors state that the restoration of splicing and full-length receptor likely increases serotonin signalling. Testing serotonin signalling should be straightforward would strengthen the claims and impact of this manuscript.

A direct measurement of serotonin signalling in the arcuate nucleus is difficult, due to the small number of only about 1000 POMC positive cells in the arcuate nucleus. We therefore tested this question indirectly by performing the experiments in melanocortin receptor 4 knock out mice that showed no statistically significant effect. It is well established that MCR4 neurons are activated by MSH after serotonergic activity in POMC neurons. We also add to the discussion to soften the statement:

In agreement with this model, oligo#5 had no statistically significant effect in MCR4 knock out mice (Srisai et al, 2011), (Figure EV5). However, since the full-length 5HT2C heterodimerizes with other seven-transmembrane receptors (Schellekens et al, 2015; Schellekens et al, 2013), it is possible that oligo#5 affects the surface localization of unknown receptor systems, which could contribute to the anorexic response.

Minor comments:

In the abstract, it should be made clear that the injection of oligonucleotide into the third ventricle was made in mice.

This was added

The chemistry of the oligo's used in the Passini 2011 paper differed from the chemistry of the oligo's used here. It is cited that the oligo's had similar chemistry, but the differences should be directly addressed.

We clarified this sentence as follows:

Previous studies using oligonucleotides with this chemistry found that cells take up the oligonucleotide directly without adjuvants (Heemskerk et al, 2009) and related 2'-O-2-methoxyethyl oligonucleotides are taken up by neurons in the brain (Passini et al, 2011), which we also observed (Figure EV2).

In figures 3A and 3B, it is not clear what the boxes below the sequencing trace data signify. Presumably the 5 edited sites? This information should be included in the figure legend.

Sorry for this omission, we add: Letters in grey boxes indicate the editing sites (also shown in Figure 1C).

RNA1 and RNA2 are discussed in reference to Figure 5, yet the identity of these RNAs is not defined until the model in Figure 6D. The nomenclature should be modified or RNA1 and 2 defined earlier in the paper.

We added RNA1 and RNA2 to the model in Figure 6E. In addition, we modified Figure 1A, to clarify the nomenclature.

In the discussion, the authors state that they sought "substitution" therapies. This has an unclear meaning. Do they perhaps mean "alternative" therapies?

We clarified this sentence: To identify therapeutic approaches that might substitute SNORD115, we performed an oligo-walk using RNA-based oligonucleotides containing

Also in the discussion section, it is stated that the oligos show no RNaseH activity. Since RNA oligos don't have RNaseH activity, but can be degraded by RNAase H, the wording should be modified for clarity.

We clarified as follows: are taken up by cells without adjuvants and show in contrast to DNA oligonucleotides no RNaseH activity

The Materials and Methods section (incorrectly labelled Material and Methods) is lacking details about dosing for the stereotaxically treated animals. This section is likewise lacking any detailed description of the systemic injection of oligo via the catheterized carotid artery. Additionally, low doses that had no statistically significant effect (data not shown) were not disclosed at all.

We already stated for the ICV injection: and used to infuse 2 μ l (1 μ g/ μ l) of RNA nucleotides over 2 minutes.

And add to the Materials and Methods:

Animals with a catheter inserted into the common carotid artery with the catheter tip directed towards the brain were purchased from Charles River (Maryland, USA). 50-100 μ g oligonucleotide (10 μ g/ μ l) were infused over 3-5 minutes. Dosages lower than 31 μ g showed no significant effect (1, 2, 10, 20 and 30 μ g were tested).

2nd Editorial Decision

23 May 2016

Thank you for the submission of your revised manuscript to EMBO Molecular Medicine. We have now received the enclosed reports from the referees that were asked to re-assess it. As you will see the reviewers are now supportive and I am pleased to inform you that we will be able to accept your manuscript pending [... editorial] final amendments.

I look forward to reading a new revised version of your manuscript.

***** Reviewer's comments *****

Referee #1 (Comments on Novelty/Model System):

Excellent work.

Referee #1 (Remarks):

All previous concerns have been addressed in the revision.

Referee #2 (Remarks):

The manuscript has been revised in response to reviewer suggestions and is much improved. In this reviewer's opinion, it is now suitable for publication.

Corresponding Author Name: Stefan Stamm

Journal Submitted to: Embo Molecular Medicine

Manuscript Number: EMM-2015-06030-V2